# Improved Sampling Algorithms for Lévy-Itô Diffusion Models

**Vadim Popov, Assel Yermekova, Tasnima Sadekova,**
Huawei Noah's Ark Lab
`{popov.vadim1,yermekova.assel1,sadekova.tasnima1}@huawei.com`

**Artem Khrapov & Mikhail Kudinov**
Huawei Noah's Ark Lab
`khrapov.artem@huawei-partners.com, kudinov.mikhail@huawei.com`

## Abstract

Lévy-Itô denoising diffusion models relying on isotropic $\alpha$-stable noise instead of Gaussian distribution have recently been shown to improve performance of conventional diffusion models in image generation on imbalanced datasets while performing comparably in the standard settings. However, the stochastic algorithm of sampling from such models consists in solving the stochastic differential equation describing only an approximate inverse of the process of adding $\alpha$-stable noise to data which may lead to suboptimal performance. In this paper, we derive a parametric family of stochastic differential equations whose solutions have the same marginal densities as those of the forward diffusion and show that the appropriate choice of the parameter values can improve quality of the generated images when the number of reverse diffusion steps is small. Also, we demonstrate that Lévy-Itô diffusion models are applicable to diverse domains and show that a well-trained text-to-speech Lévy-Itô model may have advantages over standard diffusion models on highly imbalanced datasets.

## 1 Introduction

Denoising diffusion probabilistic models (Ho et al., 2020) are a powerful class of generative models capable of solving tasks related to various continuous domains such as natural images (Dhariwal & Nichol, 2021; Gao et al., 2023), video (Luo et al., 2023), speech (Popov et al., 2021; Chen et al., 2021) and music (Hawthorne et al., 2022) to name a few. Since they were first introduced, there have been numerous attempts to overcome some of their drawbacks such as difficulty of applying them to discrete domains (Lou et al., 2024) and their inefficiency coming from iterative sampling algorithm. The latter problem has drawn much attention of researchers which has led to many successful solutions like the frameworks of flow matching (Lipman et al., 2023), consistency models (Song et al., 2023; Song & Dhariwal, 2024) and distribution matching distillation (Yin et al., 2024b;a). At the same time, other weaknesses of diffusion models have received relatively little attention. For example, conventional diffusion models having Wiener process as their driving one show substantial quality degradation on imbalanced datasets (Qin et al., 2023), and there has been a recent attempt to address this problem by replacing Wiener process with $\alpha$-stable Lévy process (Yoon et al., 2023). Such models called Lévy-Itô denoising diffusion models (LIMs) were shown to better train on imbalanced datasets achieving, in particular, better performance on rare classes. The advantage of LIMs allowing to demonstrate the superior behaviour in this scenario consists in relying on Lévy processes with discontinuous paths and employing isotropic $\alpha$-stable noise with heavier tails than those of Gaussian distribution both at training and inference.

Yoon et al. (2023) have devised necessary techniques allowing to treat Lévy-Itô models through continuous-time formalism based on stochastic calculus similar to Song et al. (2021c) who were the first to make it for common diffusion models. Particularly, the authors introduced the notion of *fractional score function* and developed *fractional denoising score matching* technique to estimate fractional score function with a neural network. Also, they designed a forward process bearing

resemblance to Variance Preserving diffusion (Song et al., 2021c) consisting in adding isotropic $\alpha$-stable noise to data until it turns to a pure noise from the prior which, in this case, is a standard symmetric $\alpha$-stable distribution. This process can be described by the forward stochastic differential equation (SDE (Oksendal, 1992)) and has a reverse-time model described by the reverse SDE written down in terms of the fractional score function containing information about data distribution and some intractable data-dependent term skipped when solving the reverse SDE. Thus, the SDE used at inference does not lead to *exact* solution (i.e. marginal probabilities of the underlying process at every diffusion time step $t$ are different from those of the forward diffusion). Two sampling algorithms were suggested by Yoon et al. (2023): the stochastic one by solving the mentioned SDE and the deterministic one by solving an analogue of what is called *probability flow ODE* in Song et al. (2021c). In contrast with the former differential equation, the latter one relies only on the fractional score function and does not contain intractable terms we have to omit at inference so it provides exact solution. At the same time, both stochastic and deterministic sampling methods of standard diffusion models are exact (both corresponding ODE and SDE rely only on the score function) and, moreover, there exists a parametric family of sampling algorithms including the deterministic one as a particular case (Song et al., 2021a). In this paper, we bridge the gap between conventional diffusion models and LIMs and derive a similar family of SDEs leading to exact sampling and providing an option to choose amount of noise at each reverse diffusion step. This new reverse dynamics makes it possible to improve performance of Lévy-Itô models when the number of function evaluations (NFE) is limited by a small number, and, as we show through empirical studies on image generation task, this improvement does not come at the cost of samples diversity, thus keeping the main advantage of LIMs intact. Figure 1 illustrates a forward Lévy diffusion and reverse denoising Lévy diffusions for various reverse SDEs we propose.

As we mentioned, the literature on conventional diffusion models offers a wide range of possible applications that so far remain almost unexplored for LIMs. In this work, we also investigate applicability of Lévy-Itô diffusion models to speech domain. To demonstrate possible advantages of LIMs over common diffusion models, we train Lévy-Itô-based text-to-speech models on imbalanced dataset with the amount of data representing different speakers varying significantly and study capability of different models to produce speech of "rare" and "frequent" speakers.

Our main contributions are threefold:

- We derive a parametric family of reverse SDEs relying only on the fractional score function whose solutions have the same marginal densities as those of the forward SDE unlike the reverse SDE proposed in the original paper.

- We demonstrate the benefits of using these SDEs at inference in terms of generated samples quality on image generation task and verify that samples diversity does not suffer if we generate data with the proposed SDEs.

- We train a Lévy-Itô text-to-speech model on a highly imbalanced dataset and evaluate its performance for speakers with different amount of training data.

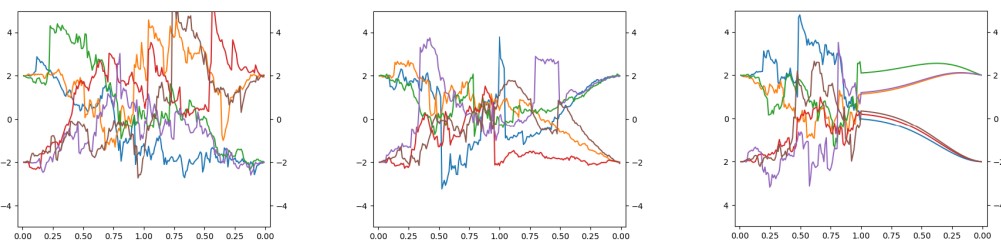

Figure 1: 1-dimensional Lévy processes with $\alpha = 1.5$. Horizontal axis stands for diffusion time $t$, vertical axis – for values of a process. Left part of each plot (time $t$ increasing from 0 to 1) is the forward diffusion. Right part (time $t$ decreasing from 1 to 0) is reverse diffusion with different amount of noise: (left) the same as in the forward diffusion ($\eta_t \equiv 1.0$); (middle) less than in the forward diffusion ($\eta_t \equiv 0.2$); (right) no noise ($\eta_t \equiv 0.0$) meaning that trajectories are continuous.

## 2 RELATED WORK

Denoising diffusion probabilistic models belong to a class of energy-based generative models (Song & Kingma, 2021). Diffusion models were first introduced as discrete-time models (Ho et al., 2020), but a more flexible continuous-time approach was proposed later (Song et al., 2021c; Song & Ermon, 2020) allowing to develop a theoretical basis under diffusion training procedure via denoising score matching (Song et al., 2021b; Hyvärinen, 2005) thanks to the fact that the forward diffusion admits reverse-time model (Anderson, 1982). Diffusion models and closely related flow matching models (Lipman et al., 2023) are very popular due to their remarkable performance in common generative tasks (Dhariwal & Nichol, 2021; Gao et al., 2023) as well as rich capabilities of controllable generation, e.g. editing an image based on various conditions (Meng et al., 2022), generating speech with the desired pitch (Sadekova et al., 2024), or putting more emphasis on input condition via classifier-free guidance (Rombach et al., 2022).

Lévy-Itô diffusion models (Yoon et al., 2023) are perhaps the most successful attempt to replace Gaussian noise in energy-based models with alternative distributions, e.g. Gamma distribution in common diffusion models (Nachmani et al., 2021), or heavy-tailed distributions in annealed Langevin dynamics (Deasy et al., 2022). Despite Lévy processes (Applebaum, 2004) have lots of applications, e.g. in finance (Eberlein, 2001) where they help to better capture dynamics of asset prices due to flexibility in choosing jump distribution (Geman, 2002), their application in score-based generative modeling is still very limited. Apart from the original paper on these models, one can mention Paquet et al. (2024) applying LIMs to protein generation, Shariatian et al. (2024) developing a discrete-time version of LIMs, and Hu et al. (2024) employing Physics-Informed Neural Networks to solve fractional partial differential equations in a more general setting than the one studied by Yoon et al. (2023).

Increasing efficiency of diffusion models based on Gaussian noise is a well-studied topic. Like the algorithm we propose in this paper for LIMs, some of these methods do not involve model fine-tuning and utilize various numerical methods of solving SDEs (Kloeden & Platen, 1992). For example, Lu et al. (2022) and Zhang & Chen (2023) make use of specific structure of reverse SDEs typically appearing in diffusion modeling to derive solvers with smaller numerical errors. Some researchers take into account both reverse and forward diffusions and design their solvers accordingly (Bao et al., 2022; Popov et al., 2022). There are solvers tailored specifically for inference with classifier guidance (Wizadwongsa & Suwajanakorn, 2023). Although methods of such kind lead to significant quality improvement for small NFE, even better efficiency can be achieved with methods requiring fine-tuning. One of the first attempts was progressive distillation (Salimans & Ho, 2022) followed by consistency modeling (Song et al., 2023; Song & Dhariwal, 2024) whose main idea is that a diffusion model performing well with a few steps should give consistent predictions of clean data on the same ODE trajectories used at inference. Distribution distillation (Yin et al., 2024b;a) involving Generative Adversarial Networks (Goodfellow et al., 2014) in model fine-tuning is a recent alternative to consistency models.

## 3 LÉVY-ITÔ MODELS

We begin this section with a quick recap of isotropic $\alpha$-stable random variables and Lévy processes as well as some basic notions from the fractional calculus. Then, a general framework of Lévy-Itô diffusion models is described.

### 3.1 ISOTROPIC $\alpha$-STABLE DISTRIBUTION

A real-valued $d$-dimensional random variable $\xi$ comes from the distribution $S\alpha S^d(\gamma)$, i.e. isotropic $\alpha$-stable distribution with scale parameter $\gamma$ and zero mean, if its characteristic function is $\mathbb{E}e^{i<u,\xi>} = e^{-\gamma^\alpha \|u\|^\alpha}$. Parameter $\alpha$ must belong to $(0, 2]$ and in general case distribution $S\alpha S^d(\gamma)$ does not have explicit form for its probability density function. A notable exception is $\alpha = 2$ in which case we have Gaussian distribution $\mathcal{N}(0, \sqrt{2}\gamma\,\mathrm{I})$ where I is $d$-dimensional identity matrix. Isotropic $\alpha$-stable distribution is a heavy-tailed one except for the case $\alpha = 2$, and, moreover, has infinite variance for $\alpha < 2$ (Paulson et al., 1975). An important property of $S\alpha S^d(\gamma)$ is that it is infinitely divisible which makes it a proper candidate to build a Lévy process upon as we discuss next.

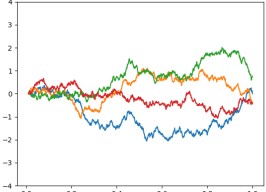 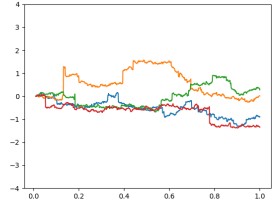 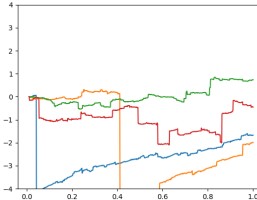

Figure 2: 1-dimensional Brownian motion (left) and Lévy processes $L_t^\alpha$ with $\alpha = 1.5$ (middle) and $\alpha = 1.2$ (right). Horizontal axis stands for time $t$, vertical axis – for values of a process. Brownian motion trajectories are almost surely continuous while trajectories of $\alpha$-stable Lévy processes are discontinuous with large jumps more probable for smaller values of $\alpha$.

## 3.2 LÉVY PROCESSES

A real-valued $d$-dimensional stochastic process $L_t$ defined for $t \geq 0$ starting at zero almost surely ($L_0 = 0$ $a.s.$) is said to be a Lévy process (Applebaum, 2004) if

- $L_t$ has stationary increments, i.e. $\text{Law}(L_{t+h} - L_h) = \text{Law}(L_t)$ for all $h > 0$;
- $L_t$ has independent increments, i.e. $L_t - L_s$ and $L_v - L_u$ are independent for all $0 \leq s < t \leq u < v$;
- $L_t$ has *càdlàg* (i.e. right continuous with left limits) paths.

This definition implies that the increments of Lévy processes must be infinitely divisible, so we can choose isotropic $\alpha$-stable distribution defined above as the one Lévy process increments belong to. In what follows we consider $\alpha$-stable Lévy processes $L_t^\alpha$ such that $\text{Law}(L_t^\alpha) = S\alpha S^d(t^{1/\alpha})$ for $\alpha \in (1, 2)$. It can be shown that, in contrast to standard Brownian motion, $\alpha$-stable Lévy processes $L_t^\alpha$ have discontinuous paths, and as $\alpha$ decreases, large jumps become more probable. These properties are illustrated in Figure 2.

Probability density functions of diffusion processes based on $L_t^\alpha$ satisfy Fokker-Planck equations, but unlike standard diffusions based on Wiener process they are not usual partial differential equations (PDEs) as Laplace operator is replaced with the fractional Laplacian (Lischke et al., 2020) when $\alpha < 2$. To be more precise, if diffusion $X_t$ satisfies the following SDE:

$$dX_t = \mu(X_{t-}, t)dt + \sigma_t dL_t^\alpha , \qquad (1)$$

where $X_{t-} = \lim_{h \to 0-} X_{t+h}$, then under certain assumptions on drift and diffusion coefficients $\mu(x, t)$ and $\sigma_t$ (Schertzer et al., 2001) probability density function $p(x, t)$ of the process $X_t$ satisfies the fractional Fokker-Planck PDE

$$\frac{\partial}{\partial t}p(x, t) = -\nabla \cdot (\mu(x, t)p(x, t)) - \sigma_t^\alpha(-\Delta)^{\alpha/2}p(x, t) \qquad (2)$$

with the proper initial conditions (Yoon et al., 2023). As one can see by comparing this fractional PDE with the common Focker-Planck PDE, the fractional Laplacian of order $\alpha/2$ denoted by $(-\Delta)^{\alpha/2}$ plays the role of (negative) Laplacian when we change diffusion driving process from standard Wiener process to $\alpha$-stable Lévy process with $\alpha < 2$.

The fractional Laplacian is a pseudo-differential operator defined through a generalization of Fourier multiplier property of the negative Laplace operator:

$$(-\Delta)^{\alpha/2} f(x) = \mathcal{F}^{-1}\{\|u\|^\alpha \mathcal{F}\{f(x)\}(u)\} , \qquad (3)$$

where $\mathcal{F}$ and $\mathcal{F}^{-1}$ denote Fourier and inverse Fourier transforms correspondingly. For the cases $-d < \alpha < 0$ and $0 < \alpha < 2$ there are alternative expressions without Fourier transforms through just a singular integral over $\mathbb{R}^d$ (Lischke et al., 2020).

## 3.3 LÉVY-ITÔ DIFFUSION MODELS

Consider adding $\alpha$-stable noise to data $X_0$ according to the forward SDE (1) on the time interval $[0, T]$ for any reasonable drift and diffusion coefficients such that for the final time $T$ we have

$\text{Law}(X_T) \approx S\alpha S^d(1)$, i.e. the prior is standard $\alpha$-stable distribution with the unit scale parameter. Yoon et al. (2023) provided a reverse-time model of a process satisfying the forward SDE (1). It is given by the SDE solved backwards in time from $t = T$ to $t = 0$ starting from $\text{Law}(X_T)$:

$$d\bar{X}_t = (\mu(\bar{X}_{t+}, t) - \alpha\sigma_t^\alpha S_t^{(\alpha)}(\bar{X}_{t+}))dt + \sigma_t d\bar{L}_t^\alpha + d\bar{Z}_t ,\tag{4}$$

where $\bar{X}_{t+} = \lim_{h\to 0+} \bar{X}_{t+h}$, $\bar{L}_t^\alpha$ is $\alpha$-stable Lévy process in reverse time (meaning that its increments $\bar{L}_s^\alpha - \bar{L}_t^\alpha$ are independent of $\bar{L}_t^\alpha$ for $s < t$) and $\bar{Z}_t$ is a reverse-time model of some data-dependent process $Z_t$ with zero mean and finite variation. The process $Z_t$ is intractable since it can be defined only through its characteristic exponent (see Theorem B.1 in Yoon et al. (2023), formula (39)) depending on data density ratios. The reverse SDE (4) is expressed in terms of the fractional score function $S_t^{(\alpha)}(x)$ defined as

$$S_t^{(\alpha)}(x) = \frac{(-\Delta)^{\frac{\alpha-2}{2}}\nabla p(x, t)}{p(x, t)} ,\tag{5}$$

where $p(x, t)$ is the probability density function of the forward diffusion (1) at time $t$.

During training the goal is to approximate this fractional score function by a neural network. Yoon et al. (2023) proposed to train score-matching neural network $s_\theta(x, t)$ with parameters $\theta$ by optimizing the fractional denoising score matching objective:

$$\mathcal{L}(\theta, t) = \mathbb{E}_{X_0, X_t}\|s_\theta(X_t, t) - S_t^{(\alpha)}(X_t|X_0)\|_2^2 ,\tag{6}$$

where conditional fractional score function is calculated using density $p(x|x_0, t)$ of conditional distribution $\text{Law}(X_t|X_0)$:

$$S_t^{(\alpha)}(x|x_0) = \frac{(-\Delta)^{\frac{\alpha-2}{2}}\nabla p(x|x_0, t)}{p(x|x_0, t)} .\tag{7}$$

If drift coefficient $\mu(x, t)$ is linear in $x$, then it can be shown that for some functions $a_t$ and $\gamma_t$ conditional distribution $X_t|X_0$ is the same as $a_t X_0 + S\alpha S^d(\gamma_t)$ which allows to compute training target explicitly:

$$S_t^{(\alpha)}(x|x_0) = -\frac{x - a_t x_0}{\alpha\gamma_t^\alpha} .\tag{8}$$

There are two ways of sampling from the trained Lévy-Itô diffusion model: either by solving the SDE

$$d\bar{X}_t = (\mu(\bar{X}_{t+}, t) - \alpha\sigma_t^\alpha S_t^{(\alpha)}(\bar{X}_{t+}))dt + \sigma_t d\bar{L}_t^\alpha ,\tag{9}$$

or by solving the probability ODE

$$dX_t = (\mu(X_t, t) - \sigma_t^\alpha S_t^{(\alpha)}(X_t))dt\tag{10}$$

with the fractional score function replaced with its neural network approximation. Both differential equations should be solved backwards in time starting from the prior $S\alpha S^d(1)$. As for the SDE (9), it is an approximate version of the exact reverse SDE (4). Yoon et al. (2023) justified omitting $d\bar{Z}_t$ term by the fact that $Z_t$ is a random process with zero mean and finite variation while $\bar{L}_t^\alpha$ has infinite variation. As far as the ODE (10) is concerned, it leads to exact sampling procedure since marginal probabilities of the solutions of (10) and (1) are the same as proven in Yoon et al. (2023).

## 4 PARAMETRIC REVERSE-TIME SDE

Conventional diffusion models allow for a parametric family of sampling algorithms (Song et al., 2021a; Kong & Ping, 2021). The parameter in these algorithms stands for amount of noise added at each inference step, and both the probability flow ODE and the reverse-time SDE are particular cases obtained for specific parameter values. The following theorem provides similar result for Lévy-Itô diffusion models:

**Theorem 1.** *Consider a stochastic process $X_t$ with marginal probability densities $p(x, t)$ driven by $\alpha$-stable Lévy process $L_t^\alpha$ and described by the forward SDE (1). Under certain regularity assumptions on drift and diffusion coefficients $\mu(x, t)$ and $\sigma_t$, non-negative time-dependent function*

$\eta_t$ and the fractional score function $S_t^{(\alpha)}(x)$ given by the formula (5) the following reverse SDE driven by reverse-time $\alpha$-stable Lévy process $\bar{L}_t^\alpha$

$$d\bar{X}_t = (\mu(\bar{X}_{t+}, t) - (1 + \eta_t)\sigma_t^\alpha S_t^{(\alpha)}(\bar{X}_{t+}))dt + \sigma_t \eta_t^{1/\alpha} d\bar{L}_t^\alpha \qquad (11)$$

has the solution with the same marginal probability densities $p(x, t)$ as the forward process $X_t$ given that its starting point $\bar{X}_T$ has the same distribution as $X_T$.

The proof of this theorem essentially relies on inspecting fractional Fokker-Planck PDEs for the SDEs (1) and (11) and can be found in Appendix A as well as a brief discussion on the necessary assumptions on $\mu(x, t)$, $\sigma_t$, $\eta_t$ and $S_t^\alpha(x)$. Note that these assumptions are basically the necessary conditions for the Fokker-Planck equations associated with the SDEs (1) and (11) to have unique solutions (Schertzer et al., 2001; Albeverio et al., 2010; Zhang et al., 2020), and it is quite common to make such kind of assumptions not only when studying diffusion generative models (Song et al., 2021b; Yoon et al., 2023), but also when studying reverse-time models of stochastic processes in general (Anderson, 1982).

Once we have trained the fractional score-matching network $s_\theta(x, t)$, we can generate samples by solving the SDE (11) backwards in time starting from the prior. The parameter $\eta_t$ controls amount of $\alpha$-stable noise added at each reverse diffusion time step $t$ and $\eta_t^{1/\alpha}$ can be thought of as the ratio between the amounts of noise in the forward and reverse diffusions given by (1) and (11): when $\eta_t \equiv 1$, then we add exactly the same amount of noise as in the forward diffusion, for $\eta_t < 1$ we add less noise which in some cases may be useful, and when $\eta_t \equiv 0$ we do not add any noise, in which case the SDE (11) coincides with the ODE (10) and its trajectories become continuous (since all jumps come solely from $\alpha$-stable noise). Figure 1 illustrates this argument.

It is worth noting that, in contrast to the SDE (4), the reverse SDEs (11) do not provide a reverse-time model of the forward process $X_t$ because distributions of the trajectories of (1) and (11) are different in general as seen from Figure 1. The main advantage of these SDEs is that they do not rely on intractable terms and can be used readily at inference providing exact marginal densities for each time $t$ given that the fractional score-matching network is trained till optimality. As for the SDE (4), we cannot use it directly and have to use the approximate SDE (9) instead.

The approximation (9) of the SDE (4) is supposed to work well since it differs by a term $d\bar{Z}_t$ where $\bar{Z}_t$ is the inverse of the process $Z_t$ with zero mean and finite variation which is negligible since $\alpha$-stable Lévy process has infinite variation. However, this argument may not work well when we use small NFE to sample from a Lévy-Itô model.

Finite or infinite variation is a property of the paths of a random process $Y_t$ defined by whether $\sup_\Pi \sum_k |Y_{t_{k+1}} - Y_{t_k}|$ over *all* partitions $\Pi$ of the interval $[0, T]$ (i.e. $0 = t_0 < t_1 < ... < t_{n-1} < t_n = T$) is finite or infinite almost surely. In practice, when we solve SDEs on a partition with relatively large step sizes, the variation of a trajectory may be far less than its supremum which may possibly be reached on partitions with smaller step sizes. In this case, the process with finite variation may have comparable contribution to the overall variation of paths as the one with infinite variation. It can be illustrated by a simple example. Let us consider

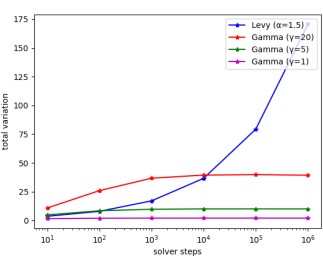

Figure 3: Variation of $L_t^\alpha$ for $\alpha = 1.5$ and $\bar{\Gamma}_t^\gamma$ for $\gamma = 1, 5$ and $20$ on $[0, 1]$ depending on number of solver steps $n$.

two processes: a process of infinite variation $L_t^\alpha$ and compensated Gamma process $\bar{\Gamma}_t^\gamma = \Gamma_t^\gamma - \gamma t$ with shape parameter $\gamma$ known to have finite variation. A linear term is subtracted from standard Gamma process $\Gamma_t^\gamma$ because we are interested in processes with zero mean. For each process $Y_t$ we can simulate its variation $\sum_k |Y_{t_{k+1}} - Y_{t_k}|$ on the interval $[0, 1]$ divided by $n + 1$ points $t_k = k/n$ into $n$ equal parts and average this value over a number of Monte-Carlo simulations. The results of this experiment are demonstrated in Figure 3. As $n$ increases, variation of $L_t^\alpha$ rapidly grows larger since it is the process of infinite variation while variation of $\bar{\Gamma}_t^\gamma$ tends to some constant. But as for small values of the number of solver steps $n$, variation of these processes are of the same order, and variation of $\bar{\Gamma}_t^\gamma$ can be even bigger. Therefore, if we want to simulate a sum of a process with finite variation and the one with infinite variation on some interval with a small number of steps, we cannot neglect increments of the process of finite variation.

Table 1: Evaluation of models trained on CIFAR10 with $\alpha = 1.8, 1.5$ and $1.2$ in terms of FID. Euler-Maruyama and Exponential Integrator solvers with $N$ steps are used.

| | Euler-Maruyama | | Exponential Integrator | | |
|---|---|---|---|---|---|
| | $N$=20 | $N$=50 | $N$=20 | $N$=50 | $N$=500 |
| SDE-A (LIM with $\alpha = 1.8$) | 144.7 | 61.57 | 10.42 | 6.58 | **2.64** |
| SDE-E (LIM with $\alpha = 1.8$) | **8.79** | **4.14** | **6.86** | **4.87** | 3.36 |
| ODE (LIM with $\alpha = 1.8$) | 11.68 | 5.23 | 10.31 | 4.88 | 3.38 |
| SDE-A (LIM with $\alpha = 1.5$) | 109.5 | 22.68 | 9.05 | 4.25 | **2.72** |
| SDE-E (LIM with $\alpha = 1.5$) | **7.86** | **4.37** | **6.27** | **4.09** | 3.26 |
| ODE (LIM with $\alpha = 1.5$) | 9.95 | 5.94 | 10.50 | 5.14 | 3.35 |
| SDE-A (LIM with $\alpha = 1.2$) | 49.87 | 4.54 | 7.70 | 4.49 | **3.28** |
| SDE-E (LIM with $\alpha = 1.2$) | **7.08** | **4.22** | **7.08** | **4.26** | 3.74 |
| ODE (LIM with $\alpha = 1.2$) | 8.43 | 5.25 | 9.14 | 5.12 | 3.74 |

Table 2: Evaluation of models trained on CIFAR10 with $\alpha = 1.8, 1.5$ and $1.2$ in terms of coverage. Euler-Maruyama and Exponential Integrator solvers with $N$ steps are used.

| | Euler-Maruyama | | Exponential Integrator | | |
|---|---|---|---|---|---|
| | $N$=20 | $N$=50 | $N$=20 | $N$=50 | $N$=500 |
| SDE-A (LIM with $\alpha = 1.8$) | 2.26% | 25.08% | 82.56% | 89.55% | **93.27%** |
| SDE-E (LIM with $\alpha = 1.8$) | **84.73%** | **90.85%** | **86.26%** | **91.26%** | 91.46% |
| ODE (LIM with $\alpha = 1.8$) | 82.36% | 89.29% | 80.35% | 89.16% | 91.05% |
| SDE-A (LIM with $\alpha = 1.5$) | 6.53% | 68.67% | 85.11% | **90.82%** | **92.63%** |
| SDE-E (LIM with $\alpha = 1.5$) | **85.39%** | **90.00%** | **87.88%** | 90.24% | 91.50% |
| ODE (LIM with $\alpha = 1.5$) | 82.72% | 88.68% | 80.93% | 88.32% | 90.73% |
| SDE-A (LIM with $\alpha = 1.2$) | 40.99% | 89.33% | 85.92% | **90.13%** | **90.74%** |
| SDE-E (LIM with $\alpha = 1.2$) | **86.01%** | **89.77%** | **86.63%** | 89.78% | 89.92% |
| ODE (LIM with $\alpha = 1.2$) | 84.07% | 88.71% | 81.26% | 87.72% | 89.95% |

To sum up, the exact reverse dynamics (11) we propose can outperform the stochastic sampling algorithm (9) when employing solvers with a small NFE because in this case the error coming from dropping $d\bar{Z}_t$ term may start to dominate. As for advantages over the deterministic sampling algorithm (10), the dynamics we propose allows to control amount of noise at inference thus potentially leading to more diverse samples for appropriate choices of $\eta_t$. We will experimentally demonstrate these advantages in the next section.

## 5 EXPERIMENTS

In this section we present the results of our experiments on image and speech modalities.

### 5.1 IMAGE GENERATION

We experiment with unconditional image generation precisely following the setting in Yoon et al. (2023). We train 3 Lévy-Itô models with $\alpha = 1.8, 1.5$ and $1.2$ on CIFAR10 with the same architecture as in the mentioned paper and evaluate different sampling schemes in terms of Fréchet Inception Distance (FID) and diversity as measured by the metric called "coverage" (Naeem et al., 2020). Loosely speaking, this metric is a probability that among $k$ nearest neighbours (as measured by the same features used to calculate FID) of a real image there exists at least one generated image. We generate $50k$ images with each model and compare them with $50k$ CIFAR10 images from the training set. We use $k = 5$ to calculate coverage. The results are shown in Tables 1 and 2. Training details can be found in Appendix B.

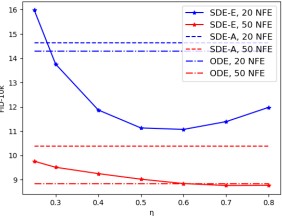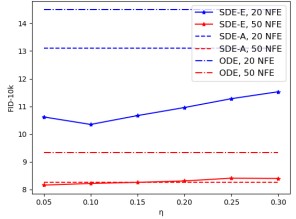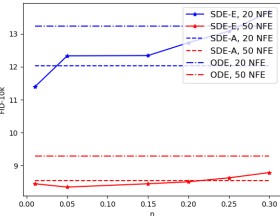

Figure 4: FID on CIFAR10 test set as a function of $\eta$ for LIMs with $\alpha = 1.8$ (left), $\alpha = 1.5$ (middle) and $\alpha = 1.2$ (right).

Two solvers with different number of steps are used to generate samples: standard Euler-Maruyama (Kloeden & Platen, 1992) (it becomes a fixed-step Euler method when we test deterministic sampling (10)) and Exponential Integrator (Zhang & Chen, 2023) (reducing to DDIM scheme (Song et al., 2021a) in a deterministic case). The latter one leads to smaller numerical errors by employing the fact that diffusions are always designed such that the term $\mu(x, t)dt$ in corresponding differential equations is linear in $x$. We test the deterministic mode according to the ODE (10) which we refer to just as *ODE*, the approximate stochastic dynamics following the SDE (9) denoted by *SDE-A*, and the exact SDE we propose (11) referred to as *SDE-E* with the parameters $\eta_t$ chosen as showing the best performance in terms of FID on CIFAR10 test set containing $10k$ images. Table 1 demonstrates that our SDE-E is the best in terms of FID for small number $N$ of solver steps, and for configurations with rather large numerical errors (i.e. Euler-Maruyama with $N = 20$ and $50$ and Exponential Integrator with $N = 20$) performance improvement is quite significant (up to $3.5$ FID) with the results being consistent for all values of $\alpha$. At the same time, the improvement becomes negligible or disappears for configurations with relatively small numerical errors. As far as generated images diversity is concerned, we observe from Table 2 that the scheme we propose is always better than the deterministic sampling (except for the case $\alpha = 1.2$ and Exponential Integrator with $N = 500$ where the difference is less than $0.05\%$) and also better than sampling from the approximate SDE for schemes with large numerical errors. These observations are consistent with the argument in Section 4.

As for tuning hyperparameters $\eta_t$, we use the following heuristics: in general, we should add very little or no noise at later stages of inference which is aligned with common practice for conventional diffusion models (Karras et al., 2022) while at earlier stages amount of noise should be considerable enough to allow for samples diversity. Therefore, we test functions $\eta_t$ that most of the time (except for some neighbourhood of 0 and $T$) equal some constant $\eta$, decrease as $t \to 0$ and increase as $t \to T$. Explicit expression for functions $\eta_t$ used for every model can be found in Appendix B. Figure 4 shows performance of different models and different solvers depending on $\eta$. One can note that typical optimal values of $\eta$ are smaller for small values of $\alpha$, i.e. the optimal overall amount of $\alpha$-stable noise added at inference should be less for noise with heavier tails.

We also conducted an experiment of training LIM with $\alpha = 1.8$ on imbalanced CIFAR10 to verify that the sampling method we propose is applicable in this scenario as well. We generated images with 20 steps of Exponential Integrator in 5 different experiment runs, and the average results are presented in Table 3. Based on these

Table 3: Performance on imbalanced CIFAR10.

| Metric | SDE-A | SDE-E | ODE |
|---|---|---|---|
| FID | 18.44 | **18.10** | 23.68 |
| Coverage | 80.33% | **81.83%** | 71.73% |

results we conclude that the sampling algorithm we propose still outperforms baseline methods both in terms of FID and diversity when training dataset is extremely imbalanced (the largest class contains 5000 images while the smallest – only 50; refer to Appendix B for more details), although performance gain is less than for models trained on common CIFAR10 dataset. It is also worth mentioning that all of the considered sampling methods corresponding to Lévy-Itô diffusion models achieve significantly better results than common diffusion models with only 60 FID in the described imbalanced setting as reported in Yoon et al. (2023).

In Table 4 we additionally report coverage for each particular class. Since Table 3 reports percentage of *all* real images from the training set having generated ones among their nearest neighbours, its

Table 4: Evaluation of the model trained on imbalanced CIFAR10 in terms of coverage measured for each class. The model generated samples with Exponential Integrator with 20 steps. Classes with larger id have less training data.

| Class id | 0 | 1 | 2 | 3 | 4 | 5 | 6 | 7 | 8 | 9 |
|---|---|---|---|---|---|---|---|---|---|---|
| SDE-A | 82% | **88**% | 77% | **82**% | 89% | **68**% | 81% | 65% | 80% | **74**% |
| SDE-E | **84**% | 87% | **82**% | 81% | **92**% | 65% | 84% | **69**% | **91**% | 70% |
| ODE | 73% | 83% | 63% | 73% | 80% | 58% | **87**% | 55% | 86% | 71% |

Table 5: Evaluation of text-to-speech models trained with Gaussian noise and $\alpha$-stable noise for $\alpha = 1.8$ and $1.5$ in terms of speaker similarity. Euler method of solving ODE with $N$ steps is used.

| Female speaker (1000 min.) | $N = 30$ | $N = 50$ | $N = 100$ |
|---|---|---|---|
| Gaussian noise | $0.815 \pm 0.011$ | $0.834 \pm 0.010$ | $0.850 \pm 0.010$ |
| $\alpha$-stable noise ($\alpha = 1.8$) | $0.822 \pm 0.010$ | $0.843 \pm 0.009$ | $0.853 \pm 0.010$ |
| $\alpha$-stable noise ($\alpha = 1.5$) | $\mathbf{0.841 \pm 0.009}$ | $\mathbf{0.859 \pm 0.009}$ | $\mathbf{0.867 \pm 0.009}$ |
| Male speaker (10 min.) | $N = 30$ | $N = 50$ | $N = 100$ |
| Gaussian noise | $0.738 \pm 0.009$ | $0.762 \pm 0.010$ | $0.773 \pm 0.010$ |
| $\alpha$-stable noise ($\alpha = 1.8$) | $0.761 \pm 0.009$ | $0.780 \pm 0.010$ | $0.794 \pm 0.010$ |
| $\alpha$-stable noise ($\alpha = 1.5$) | $\mathbf{0.782 \pm 0.009}$ | $\mathbf{0.792 \pm 0.008}$ | $\mathbf{0.800 \pm 0.009}$ |

results are strongly biased towards more represented classes, and, in principle, it could have happened so that the coverage gain of $1.5\%$ achieved by our method SDE-E holds mostly for such classes while its coverage for less represented classes is poor. Table 4 shows that this is not the case: one cannot observe dramatic quality drop in terms of coverage of SDE-E compared to SDE-A for classes containing small number of images.

## 5.2 Speech synthesis

In experiments with LIMs on text-to-speech task we closely follow Popov et al. (2021) and train models with the text encoder, duration predictor and diffusion-based decoder for three kinds of driving noise: Gaussian as in the original paper and $\alpha$-stable with $\alpha = 1.8$ and $1.5$. To demonstrate benefits of utilizing heavy-tailed noise distribution, we train text-to-speech models on extremely imbalanced dataset consisting of 16.6 hours (1000 minutes) of an English female speaker (Ito, 2017) and 10 minutes of an English male speaker with id 9017 from Bakhturina et al. (2021). Appendix C contains training details.

We synthesized 100 sentences for each speaker using $30, 50$ and $100$ ODE solver steps. CAM++ speaker verification model (Wang et al., 2023) was chosen to evaluate speaker similarity to ground-truth recordings. Table 5 shows the results of this evaluation. It is clear that the models with $\alpha$-stable noise consistently outperform the one based on Gaussian noise for both speakers in terms of speaker similarity, and the difference is more distinct when a small number of steps is used for generation. Furthermore, the model with $\alpha = 1.5$ shows better results than the one with $\alpha = 1.8$. However, for all models average speaker similarity values are lower for the less represented speaker. We also have to note that training on such imbalanced dataset was a challenging task for all models under comparison: we observed that speech generated with male speaker's voice had quite a lot of inaccuracies in pronunciation irrespective of the model.

## 6 Limitation

One of the main limitations of the reverse dynamics we derive in the paper is that we have to tune parameters $\eta_t$ corresponding to amount of noise added at inference which may be quite time-consuming. There are papers on estimating optimal reverse variance in common diffusion models (Bao et al., 2022), but in case of $\alpha$-stable Lévy processes we have infinite variance, therefore we cannot apply such methods directly. So, finding an optimal amount of $\alpha$-stable noise at inference is

an interesting open question. Estimating increments $d\bar{Z}_t$ of a process of finite variation skipped in the original reverse dynamics may also be of interest and lead to alternative sampling algorithms. As far as text-to-speech experiments are concerned, it could be useful to focus on more realistic scenarios when we have several "rare" speakers rather than only one.

## 7 CONCLUSION

In this paper we propose a parametric family of algorithms of sampling from Lévy-Itô diffusion models relying on the reverse-time SDE. This SDE is derived to have solutions with the same marginal probabilities as those of the forward diffusion which makes the proposed sampling algorithms valid from theoretical point of view. Our approach is tested on image generation task where it is shown to significantly improve the results of unconditional image generation when using a small number of generation steps without sacrificing samples diversity. Besides, we study applicability of Lévy-Itô diffusion models to speech domain and demonstrate the potential of text-to-speech Lévy-Itô models in the scenario when multi-speaker training data is highly imbalanced.

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

## A  PROOF OF THEOREM 1

We consider the following forward SDE

$$dX_t = \mu(X_{t-}, t)dt + \sigma_t dL_t^\alpha \tag{12}$$

and reverse SDE

$$d\bar{X}_t = (\mu(\bar{X}_{t+}, t) - (1 + \eta_t)\sigma_t^\alpha S_t^{(\alpha)}(\bar{X}_{t+}))dt + \sigma_t \eta_t^{1/\alpha} d\bar{L}_t^\alpha \tag{13}$$

on the time interval $[0, T]$. $L_t^\alpha$ and $\bar{L}_t^\alpha$ are forward and reverse-time $\alpha$-stable Lévy processes correspondingly, and the fractional score function $S_t^{(\alpha)}(x)$ is defined as

$$S_t^{(\alpha)}(x) = \frac{(-\Delta)^{\frac{\alpha-2}{2}}\nabla p(x,t)}{p(x,t)} \,, \tag{14}$$

where $p(x,t)$ is the marginal density of the forward diffusion $X_t$. We assume that both (12) and (13) have solutions and $p(x,t)$ is the unique solution of the fractional Fokker-Planck equation associated with the forward diffusion (12)

$$\frac{\partial}{\partial t}p(x,t) = -\nabla \cdot (\mu(x,t)p(x,t)) - \sigma_t^\alpha(-\Delta)^{\alpha/2}p(x,t) \tag{15}$$

with the proper initial conditions. We also assume that the fractional Fokker-Planck equation associated with the reverse diffusion (13) has unique solution. The necessary conditions include measurability of $\mu(\cdot, t)$ and $S_t^{(\alpha)}(\cdot)$ and certain Lipschitz-type constraints on drift and diffusion coefficients of the SDEs (12) and (13). More details can be found e.g. in Albeverio et al. (2010); Schertzer et al. (2001); Zhang et al. (2020).

*Proof.* Definition of the fractional Laplacian (3) and Fourier multiplier property of negative Laplace operator $\Delta$ imply that

$$(-\Delta)^{\frac{\alpha-2}{2}}(-\Delta f(x)) = \mathcal{F}^{-1}\left\{\|u\|^{\alpha-2}\mathcal{F}\left\{\mathcal{F}^{-1}\left\{\|u\|^2\mathcal{F}\{f(x)\}\right\}\right\}\right\}$$
$$= \mathcal{F}^{-1}\{\|u\|^\alpha\mathcal{F}\{f(x)\}\} = (-\Delta)^{\frac{\alpha}{2}}f(x) \,, \tag{16}$$

which makes definition of the fractional Laplacian consistent with the standard Laplace operator.

Now we want to make a reverse-time process $\bar{X}_t$ a forward-time one and write down the fractional Fokker-Planck equation associated with it. Consider a change of time variable $\tau = T - t$. Put $Y_\tau = \bar{X}_t$ and rewrite (13) using $dt = -d\tau$:

$$dY_\tau = -(\mu(Y_{\tau-}, T-\tau) - (1 + \eta_{T-\tau})\sigma_{T-\tau}^\alpha S_{T-\tau}^{(\alpha)}(Y_{\tau-}))d\tau + \sigma_{T-\tau}\eta_{T-\tau}^{1/\alpha}dL_\tau^\alpha \,. \tag{17}$$

The SDE (17) is now a forward one (time $\tau$ flows forward from 0 to $T$), and its fractional Fokker-Planck equation is

$$\frac{\partial}{\partial\tau}q(x,\tau) = \nabla\cdot\left(\left(\mu(x,T-\tau) - (1+\eta_{T-\tau})\sigma_{T-\tau}^\alpha S_{T-\tau}^{(\alpha)}(x)\right)q(x,\tau)\right) - \eta_{T-\tau}\sigma_{T-\tau}^\alpha(-\Delta)^{\frac{\alpha}{2}}q(x,\tau) \tag{18}$$

where $q(x,\tau)$ is the probability density function of the process $Y_\tau$.

Let us also rewrite the equation (15) in terms of $\tau$ using $\frac{\partial}{\partial t} = -\frac{\partial}{\partial\tau}$:

$$\frac{\partial}{\partial\tau}p(x,T-\tau) = \nabla\cdot(\mu(x,T-\tau)p(x,T-\tau)) + \sigma_{T-\tau}^\alpha(-\Delta)^{\alpha/2}p(x,T-\tau) \,. \tag{19}$$

We aim to prove that $Y_\tau$ and $X_t$ have the same marginal densities, i.e. that $q(x,\tau) = p(x,t) = p(x,T-\tau)$. By our assumption equation (18) has unique solution, so it is sufficient to check that $p(x,T-\tau)$ satisfies this equation.

The fractional Laplacian property (16) implies that

$$-\nabla\cdot\left(p(x,T-\tau)S_{T-\tau}^{(\alpha)}(x)\right) = -\nabla\cdot\left(p(x,T-\tau)\frac{(-\Delta)^{\frac{\alpha-2}{2}}\nabla p(x,T-\tau)}{p(x,T-\tau)}\right)$$
$$= -\nabla\cdot\left((-\Delta)^{\frac{\alpha-2}{2}}\nabla p(x,T-\tau)\right) = (-\Delta)^{\frac{\alpha-2}{2}}(-\Delta p(x,T-\tau)) = (-\Delta)^{\frac{\alpha}{2}}p(x,T-\tau) \,. \tag{20}$$

Substituting $q(x, \tau)$ with $p(x, T - \tau)$ in (18) and employing (20) we have

$$
\begin{aligned}
\frac{\partial}{\partial \tau} p(x, T - \tau) &= \nabla \cdot (\mu(x, T - \tau) p(x, T - \tau)) - (1 + \eta_{T-\tau}) \sigma_{T-\tau}^{\alpha} \nabla \cdot \left( S_{T-\tau}^{(\alpha)}(x) p(x, T - \tau) \right) \\
&\quad - \eta_{T-\tau} \sigma_{T-\tau}^{\alpha} (-\Delta)^{\frac{\alpha}{2}} p(x, T - \tau) = \nabla \cdot (\mu(x, T - \tau) p(x, T - \tau)) \\
&\quad + (1 + \eta_{T-\tau}) \sigma_{T-\tau}^{\alpha} (-\Delta)^{\frac{\alpha}{2}} p(x, T - \tau) - \eta_{T-\tau} \sigma_{T-\tau}^{\alpha} (-\Delta)^{\frac{\alpha}{2}} p(x, T - \tau) \\
&= \nabla \cdot (\mu(x, T - \tau) p(x, T - \tau)) + \sigma_{T-\tau}^{\alpha} (-\Delta)^{\frac{\alpha}{2}} p(x, T - \tau)
\end{aligned}
\tag{21}
$$

which is satisfied by $p(x, t)$ due to (19).

$\square$

## B  ADDITIONAL DETAILS OF EXPERIMENTS IN IMAGE GENERATION

The model we use for CIFAR10 experiments is *NCSN++(deep)* (Yoon et al., 2023; Song et al., 2021c) with 8 residual blocks. We train 3 models for $\alpha = 1.8$, $1.5$ and $1.2$ with batch size 128 and learning rate $0.0001$ for $250k$ iterations. Diffusion models tend to overfit on CIFAR10 so we choose the best checkpoint in terms of FID on the test set ($100k$, $150k$ and $180k$ iterations for $\alpha = 1.8$, $1.5$ and $1.2$ respectively).

As for tuning hyperparameters $\eta_t$, we choose the following expression

$$
\eta_t = \eta + (\delta - \eta) \left[ \frac{t - t_0}{T - t_0} \right]^{+} + 2\eta(\sigma(ct) - 1) ,
\tag{22}
$$

where $[x]^{+} = \max\{0, x\}$ and $\delta \geq \eta$. Diffusion models we train are designed to have $T = 1$. The function (22) decays fast towards zero as $t \to 0$ because of the last sigmoid term and linearly increases from $\eta$ at time step $t = t_0$ to $\delta$ at time step $t = T$. We choose $c = 20$ in all experiments to ensure fast decay in the neighbourhood of $t = 0$. We take $t_0 = 0.7$, $0.85$ and $0.95$ for $\alpha = 1.8$, $1.5$ and $1.2$ correspondingly. $\delta$ is always chosen to be either $1$ or $\eta$ (meaning no increase in the neighbourhood of $t = T$). Optimal values $(\eta, \delta)$ we choose to calculate FID on CIFAR10 train set are given in Table 6.

Table 6: Hyperparameters for Euler-Maruyama (EM) and Exponential Integrator (EI) with different number of steps (20, 50 or 500).

| $(\eta, \delta)$ | EM-20 | EM-50 | EI-20 | EI-50 | EI-500 |
|---|---|---|---|---|---|
| $\alpha = 1.8$ | $(0.10, 0.10)$ | $(0.25, 0.25)$ | $(0.60, 1.00)$ | $(0.70, 1.00)$ | $(0.10, 0.10)$ |
| $\alpha = 1.5$ | $(0.10, 0.10)$ | $(0.15, 1.00)$ | $(0.10, 1.00)$ | $(0.05, 1.00)$ | $(0.10, 0.10)$ |
| $\alpha = 1.2$ | $(0.15, 0.15)$ | $(0.10, 1.00)$ | $(0.01, 1.00)$ | $(0.10, 1.00)$ | $(0.10, 0.10)$ |

Typical shapes of the function $\eta_t$ as well images generated by the LIM with $\alpha = 1.8$ and Exponential Integrator solver with 20 steps are given in Figure 5.

Imbalanced CIFAR10 contained 5000, 2997, 1796, 1077, 645, 387, 232, 139, 83 and 50 images belonging to classes "airplane", "automobile", "bird", "cat", "deer", "dog", "frog", "horse", "ship" and "truck" correspondingly. It is the same setting as that used in Yoon et al. (2023). For the model trained on this dataset we chose $\eta_t = t^{1/5}$.

## C  ADDITIONAL DETAILS OF EXPERIMENTS IN SPEECH SYNTHESIS

In speech synthesis experiments we worked with the model closely following Popov et al. (2021) except for several modifications. First, the forward diffusion transforms data distribution into $\mathcal{N}(0, I)$ instead of $\mathcal{N}(\mu, I)$ in the baseline model and into $S\alpha S^d(1)$ in case of Lévy-Itô model. Second, early experiments revealed the fact that training on imbalanced dataset introduces a bias in the text encoder

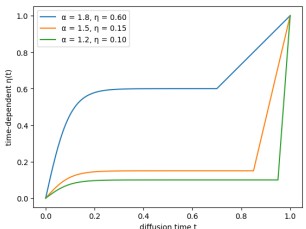 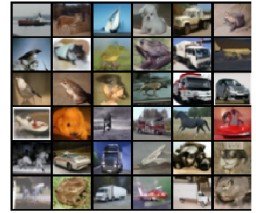 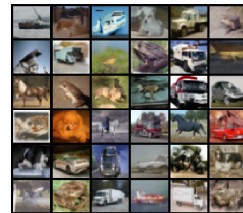

Figure 5: Typical shapes of $\eta_t$ (left) and images generated with the approximate SDE (9) (middle) and the exact one (11) we propose (right) with 20 iterations of Exponential Integrator.

towards the voice with the highest ratio. To avoid this issue we replaced this block with the separately trained text encoder which predicts "average voice" mel-spectrograms as in Sadekova et al. (2022). These features are speaker-independent and preserve linguistic content, so speaker information is encoded by the diffusion decoder only.

The encoder consists of a stack of three convolutional layers with kernel size 5 and 512 channels followed by a bidirectional LSTM with 256 units and the duration predictor with an upsampling module. Duration predictor is a two-layer bidirectional LSTM with 256 units. The whole block is pre-trained on LibriTTS dataset, fixed and works in the teacher-forcing mode during the decoder training which took 2800 epochs. Ground truth phoneme durations are obtained with Montreal Forced Aligner. The pre-trained universal HiFi-GAN (Kong et al., 2020) is used as a vocoder.

