# OpenReview forum: "Improved Sampling Algorithms for Lévy-Itô Diffusion Models"
_ICLR.cc/2025/Conference — ICLR 2025 Poster_

### Official Review · Reviewer_fdtf · 2024-10-25

**Soundness:** 3
**Presentation:** 2
**Contribution:** 3
**Rating:** 5
**Confidence:** 2

**Summary:**

A family of inverse dynamics is derived for diffusion models that use isotropic (\alpha)-stable noise instead of Gaussian noise. Such models have been proposed recently with a deterministic inverse dynamics and a stochastic one. The deterministic dynamics is guaranteed to retrieve the exact marginal distributions. Unfortunately, the stochastic dynamics yields only approximate solutions.  This paper addresses this limitation. The algorithm is expected to be effective for data generation on imbalanced datasets. Its practical usefulness is experimentally examined in applications to image generation and text-to-speech tasks.

**Strengths:**

A parametric family of inverse dynamics is derived for diffusion models that use isotropic (\alpha)-stable noise instead of Gaussian noise. Unlike the previously proposed one, this family is guaranteed to exactly retrieve the target marginal distributions for any parameter setting in ideal cases.

**Weaknesses:**

Writing in section 3.3 is sloppy. Since "some intractable data-dependent process Z_t" is not explained sufficiently, it is difficult to grasp the relation between the previously proposed algorithm and the current one.

**Questions:**

The relation between the algorithm by Yoon et al (2023) and the current one is unclear. Please explain more details on how equations (4) and (11) are related.

---

> ### Author Response · Authors · 2024-11-19
> **Reply**
>
> We would like to thank the reviewer and take the opportunity to explain the place in our paper that was unclear to them.
>
> The process $Z_t$ can be defined only through its characteristic exponent depending on the ratio of corrupted data densities, that's why we refer to this process as "intractable data-dependent process". The precise formula of the characteristic exponent of this process can be found in Appendix B of the paper [1] it has first been introduced in (Theorem B.1, formula (39)). This process has zero mean and is of finite variation as we mention in our paper, and one can see from the formula (39) in [1] that it is indeed hard to characterize it more precisely without writing down this formula. We added reference to it in the revised version of our paper.
>
> The SDE (4) proposed in the original paper on LIMs [1] and the SDE (11) we propose in our paper have the following key properties allowing to compare them:
> - Both SDEs are exact, which means that marginal densities of their solutions are equal to those of the forward SDE (1), i.e. $p(\bar{X}_t^{(4)})=p(\bar{X}_t^{(11)})=p(X_t^{(1)})$ for every $t$.
> - Moreover, the distribution of the continuous time *trajectories* of the SDE (4) in the time interval $t\in[0,T]$ is the same as that of the forward SDE (1). The same can't be said about the trajectories of the SDE (11). So, this is the advantage of the SDE (4) proposed in the seminal paper on LIMs [1].
> - The disadvantage of the SDE (4) is that it contains term $d\bar{Z}_t$ that can neither be calculated analytically nor estimated by a neural network. This term is just skipped during sampling. In contrast, the SDE (11) we propose has the advantage of containing only tractable terms or the ones estimated by the score-matching neural network.
> - Another advantage of the SDE (11) we propose is that it allows for more flexible sampling since it depends on the non-negative parameter $\eta_t$ which can be tuned once the Levy-Ito diffusion model is trained. This parameter stands for the amount of noise added at each reverse diffusion step and can depend on time $t$.
>
> Thus, both SDEs have advantages and disadvantages and can have different performance in different scenarios. In Section 4 of our paper after introducing the SDE (11) we discuss the reasons why this SDE can be better suited for low NFE scenario, i.e. when we want to sample from a diffusion model by taking a small number of reverse diffusion steps. After that, we experimentally verify our hypothesis in Section 5.
>
> [1] Score-based Generative Models with Levy Processes, Yoon et al.

---

### Official Review · Reviewer_9Gsh · 2024-10-27

**Soundness:** 3
**Presentation:** 2
**Contribution:** 2
**Rating:** 6
**Confidence:** 3

**Summary:**

The paper offers a novel method for sampling using a Lévy-Itô diffusion model based on a new formulation of its corresponding SDE. The authors identify an issue in existing methods for the reverse SDE caused by neglecting one of the terms, and offer a solution. The paper includes experiments which compare the generation results of the proposed solutions to alternatives, as well as a demonstration of its use in addressing skewed training datasets in the text-to-speech field.

**Strengths:**

- The paper analyzes Lévy processed and their use for diffusion models, a field which is under-explored.
- The proposition in the paper is straightforward and mathematically justified.
- The experiments in the paper provide evidence of the benefits of the new method.

**Weaknesses:**

- The presentation of the paper could be improved, both at the sentence and at the paragraph level.
- The figures and naming scheme makes it hard to follow the result and illustrations in the paper. (specifically in Fig. (4), Fig (1), Tab. (5))
- The takeaway from the experiment shown in Tab. (4) is unclear. While Tab. (3) shows a small advantage for the proposed method, the results in Tab. (4) do not reflect that.
- The evidence for an advantage in underrepresented data could be made stronger (using more examples, specifically in the image domain).

If some of my weaknesses are properly addressed I am inclined to raise my score.

**Questions:**

- I believe it would be easier to follow the comparisons by using a consistent naming scheme, instead of the current SDE(<number>)
- How do extremely small NFEs (5-15) effect the sampling quality of the proposed method?

---

> ### Author Response · Authors · 2024-11-19
> **Reply**
>
> We would like to thank the reviewer for the comments. We revised our paper taking them into consideration (see general comment above). Below we give detailed comments.
>
> 1. Following your suggestion, we changed the notation in the experimental section of our paper. We hope that these changes will have positive effect on its readability and facilitate comparison of different models and methods. We denoted stochastic sampling method from the original paper on Levy-Ito models by SDE-A (approximate SDE, former SDE(9)), the method we propose - by SDE-E (exact SDE, former (11)), and deterministic sampling algorithm - by ODE (former ODE (10)). Tables and figures were updated accordingly.
>
> 2. We performed 4 additional experiment runs to evaluate the model trained on imbalanced CIFAR10 dataset according to the comment of the Reviewer EF2f who had concerns about small performance gap between SDE-E and SDE-A in this setting. The results averaged across 5 different experiment runs are presented in the updated Tables 3-4 (see also item 5 in the general comment about the changes in the revised version of the paper). The purpose of Table 4 is to show that our sampling method SDE-E does not lead to significant quality degradation on small classes (i.e. classes with larger id in Table 4). And this is what we observe: for some classes SDE-E gives slightly better results, for some SDE-A does, but there are no classes for which our method SDE-E shows dramatic performance drop compared to SDE-A. This is important for the following reason: since coverage in Table 3 is calculated as percentage of all real images from the training dataset having generated ones among their $k=5$ nearest neighbours, this statistics is biased towards classes with more images, and one can imagine a situation in which the overall coverage for SDE-E is larger than that of SDE-A by 1.5% (as follows from Table 3), but this gain is achieved solely because of improved diversity on large classes, while performance on less represented classes is very poor. Table 4 demonstrates that this is not the case.
>
> 3. The main experimental claim of our paper related to the new sampling algorithm for LIMs is that SDE-E leads to improved quality (in general) when a small NFE is used for generation, and this gain does not come at the cost of decreased samples diversity (in general). In image experiments, Tables 1-3 support this claim. We do not claim that our method leads to significant improvements compared to the conventional LIM sampling method SDE-A in imbalanced setting for under-represented data. However, since the capability to generate data from less-represented data modes is an important property of LIMs, we show that this capability is not lost when we generate samples using our method SDE-E. Table 4 demonstrates this claim.
>
> 4. Typically, for extremely small NFEs ODE method works better than all stochastic sampling methods. For example, for the model trained on imbalanced CIFAR10, ODE with 10 NFEs gives 73.5 FID while SDE-A gives 75.1 FID with the same NFEs. Our method SDE-E is a generalization of the deterministic method ODE (when $\eta_t=0$ for every $t$). Anyway, all known methods work not very well for extremely small NFEs and quality of the generated images having FID around 60-70 is bad. In fact, we chose 20 NFEs as the smallest number to demonstrate our findings since it is the minimal number sufficient to generate images of fair quality. We think that to make LIMs work well in extremely low NFE scenario employing optimal sampling schemes alone is not sufficient, and techniques requiring fine-tuning (e.g. Progressive Distillation or Consistency Modeling) are necessary.

---

> > ### Comment · Reviewer_9Gsh · 2024-11-26
> >
> > I thank the reviewers for their response. As some of my concerns were addressed, I have raised my score.

---

### Official Review · Reviewer_EF2f · 2024-10-29

**Soundness:** 4
**Presentation:** 3
**Contribution:** 3
**Rating:** 8
**Confidence:** 3

**Summary:**

Levy diffusion models perform better (especially on rare classes) when it’s trained on imbalanced datasets. However, the previous reverse Levy process doesn’t have same marginal probabilities at each noise level due to the omitted intractable term, resulting in non-exact sampling.
The authors propose a novel parametric family of SDE whose solutions have the same marginal densities as the forward levy diffusion process, leading to exact solutions.
Empirical experiments demonstrate that the proposed reverse process has superior sample quality with small number of sampling steps, and also performs good in terms of sample diversity.

**Strengths:**

- The paper is well-organized, and the technique derivations are solid. Also most of the arguments are well supported by experiments.
- Instead of developing a reverse-time process of the forward Levy-Ito SDE, the authors propose parametric reverse-time SDEs that have the same marginal probability densities as the forward process, which gives exact sampling.
- It seems like compare to the baseline SDE, the proposed one is good with multiple numerical solvers. For example, in Table 1 and 2, SDE(11) performs good with both solvers, while baseline SDE approximate(9) has a different performance with different solvers.

**Weaknesses:**

- Though the improvement of FID scores given small number of sampling steps (e.g. N=20) is huge, the improvement on imbalanced CIFAR10 is marginal. Also, it could be nice to provide some FID scores of samples from Gaussian diffusion models, which shows a clear improvement in imbalanced dataset.
- In the speech synthesis experiment, the authors compare the proposed model only with the Gaussian-based diffusion model. However, it remains unclear whether this model is still SOTA when compared to the baseline Lévy-Itô diffusion model.

**Questions:**

- In table 3, I wonder how many times the authors run the experiments? It seems that for both FID and coverage metrics, the proposed SDE has very close performance to the baseline SDE.
- In the speech synthesis experiment, could you provide a comparison to the baseline Lévy-Itô diffusion model?

---

> ### Author Response · Authors · 2024-11-19
> **Reply**
>
> We would like to thank the reviewer for questions and suggestions that lead to updates in the new revision of the paper (see general comment) and briefly comment on them below.
>
> 1. We added results standard Gaussian diffusion models achieve in imbalanced CIFAR10 setting showing that LIMs significantly outperform them irrelevant of the sampling method (60 FID for DMs vs 18-23 FID for LIMs).
>
> 2. In the initial version of the paper we performed one experiment run to obtain results for the model trained on imbalanced CIFAR10 dataset. Since the difference between our method (sampling from exact SDE, or SDE-E) and the conventional one (sampling from approximate SDE, or SDE-A) is quite little as you've pointed out, we decided to do 4 additional experiment runs and average the results across all 5 experiment runs to obtain more reliable results. We'd like to mention that in each of these runs SDE-E outperformed SDE-A both in FID and coverage, and the average results show that the difference even grew slightly larger (0.3 FID and 1.5% coverage) compared to the initial version with only one experiment run.
>
> 3. In speech synthesis experiments, ODE-based sampling is used for all three models. The original text-to-speech model Grad-TTS our models are based on uses deterministic ODE-based synthesis since it is preferable in low NFE regime. We think that the strong conditioning (i.e. text) is the reason why we don't need any additional uncontrolled speech variation and benefit from deterministic sampling. Performance of both SDE-A and SDE-E we propose are inferior to that of ODE when small NFE is used, and all three methods perform comparably well when the number of generation steps is large (around 1000). This is why for speech synthesis experiments we report the results of two LIMs ($\alpha=1.8$ and $\alpha=1.5$) only with the common solver for ODE (10) (which is, by the way, a particular case of the SDE (11) we propose for $\eta_t=0$).

---

> > ### Comment · Reviewer_EF2f · 2024-11-25
> > **Thanks for the response**
> >
> > Thanks for your effert, I have increased my score accordingly.
> >
> > Good luck!

---

### Official Review · Reviewer_7oeL · 2024-11-03

**Soundness:** 3
**Presentation:** 3
**Contribution:** 4
**Rating:** 8
**Confidence:** 3

**Summary:**

This paper introduces a paremetric family of SDEs allowing different sampling scheme from a single pretrained Levy Ito diffusion model. It brings the toolbox for Levy Ito diffusion models (LIM) closer to that of standard "Gaussian" diffusion models (DM). Empirically, the authors find that the new paremetric family of SDEs has benefit in term of generated samples quality and diversity at low number of functional evaluations, which lowers the computational cost of sampling from Levy Ito diffusion models. Finally, they train a text-to-speech diffusion model on an imbalanced dataset and evalutate the benefits of LIMs compared to DM.

**Strengths:**

* The parametric family of SDEs derived by the author is new and offer more flexibility for LIM sampling.
* The authors give a rigorous derivation of the proposed SDEs
* The paper clearly tries to show how their theoretical results compare with existing litterature on LIMs. It explains clearly what is the limitation of Yoon et al (approximation of the reverse process).
* Promising results for text to speech models
* The paper is well written overall

**Weaknesses:**

* The experiments about text to speech models are interesting and promising but they are not tied to the main results of the paper (theoretical and empirical contributions to LIM sampling). The baseline DM against which authors compare is also new and has the modifications outlined in lines 809 to 825 in the Appendix.
* The simple toy example explains well why processes with zero mean and finite variations cannot be omitted completely in low NFE schemes. However, in my opinion, the paper does not explain clearly what steps/changes in the parametric family of SDEs prevents any such (intractable) process to come into play. I understand from line 286 that the reverse time SDE is no longer a model of the forward time SDE and that trajectories are different. Is that what allows you to have "better" SDEs?
* (Not a real weakness per se) The new paremetric family of SDEs does not improve the capabilities of LIM in term of imbalance "correction". As examplified by table 4.

**Questions:**

* Please explain weakness 2.
* For the text to speech model, even on the main mode (female speakers), LIMs outerperform Gaussian DMs. Why is that? Wouldn't we expect Gaussians DM to perform better on the main mode of the distribution? Does fixing the "bias" in the text encoder (line 809 to 825) unfairly disadvantage Gaussians DM even on the main mode? What would the results have been with a standard encoder (or alternatively do you have other baselines for text to speech imbalanced modeling that would shed light on the benefits of LIM on both the main mode and the tail mode)? You are also using an ODE sampler, which, in your image generation experiment, favors the less frequent class.
* It seems that the common accepted reason for which LIMs are better for imbalanced datasets is that Lévy Ito processes, with their heavy tails and jump possibilities, better cover less probable isolated modes of the data distribution. In your opinion, why does this phenomenon subsist when using ODE sampling? In your imbalanced CIFAR experiment, classes 8 and 9 seem favored by the ODE.

---

> ### Author Response · Authors · 2024-11-19
> **Reply to Q1**
>
> We would like to thank the reviewer for very interesting questions and topics to discuss. In what follows we address those.
>
> Both the SDE (4) and (11) have the solutions with the same marginal likelihoods as those of the forward process $X_{t}$ given by the SDE (1). So, if fractional score matching neural network is trained till optimality, they both lead to the same exact solutions $\bar{X}_t$ such that $p(\bar{X}_t)=p(X_t)$ for all $t$, and $p(\bar{X}_0)=p(X_0)$ in particular.
>
> In practice, we generate samples from the distribution having density different from $p(X_0)$ for several reasons including:
>
> (i) errors coming from imperfect score function approximation $S_{t}^{(\alpha)}(x)$;
>
> (ii) discretization errors coming from drift and diffusion terms;
>
> (iii) in case of SDE (4), error coming from omitting $d\bar{Z}_t$ term.
>
> The errors (i) are the same for both methods in each point $x$, and the errors (ii) are comparable because diffusion term in SDE (11) is less than that in SDE (4) (since we consider $\eta_{t}\in [0,1]$) and drift terms corresponding to imperfectly estimated score function differ by a factor of $(1+\eta_{t}) / \alpha$ where $1+\eta_{t}$ is between 1 and 2 and $\alpha$ belongs to the same interval. Our hypothesis is that in low NFE regime SDE (11) leads to better results than (4) because the error (iii) associated with the SDE (4) dominates errors (ii) which are comparable for both methods.

---

> ### Author Response · Authors · 2024-11-19
> **Reply to Q2**
>
> 1. In our understanding, LIMs should outperform common Gaussian diffusion models (DMs) on both frequent and rare classes. We suppose that the main reason could be that forward diffusion trajectories of LIMs connecting prior with two distinct classes are more "disentangled" than those of DMs, and during reverse generative steps it is less probable to follow the path leading to class A when conditioned on class B on early steps having to correct the path on later steps (which harms generation quality). In this argument it seems not to matter which class, A or B, is larger. While this is only our hypothesis, there are experiments on conditional image generation on imbalanced CIFAR10LT dataset in the paper [1] (see Table 3) that reveal the same phenomenon: LIMs outperform DMs in terms of FID both on rare and frequent classes. As expected, FID scores for more frequent classes in that experiment are better which also accords with our speech experiment whose results are shown in Table 5.
>
> 2. As far as the design of text-to-speech models is concerned, we did perform experiments with the standard encoder used in the original Grad-TTS model. However, it turned out that with this design, both LIM and DM coped well with generating frequent female voice, but were completely unable to generate rare male voice - both models just ignored class label and always generated female voice. We think that it happened for the following reasons:
> - the encoder produced mel features that sounded as if pronounced by the female voice (because the encoder in this text-to-speech model is trained on target mel features with MSE loss on batches consisting mostly of female speech, the ratio was
> 100:1);
> - in the original single-speaker text-to-speech model Grad-TTS, diffusion-based decoder just refined
> "rough" mel features corresponding to a single speaker produced by the encoder;
> - in our two-speaker imbalanced setting, diffusion-based decoder essentially had just to refine the outputs of the encoder for frequent female speaker while for rare male speaker it had first to change timbre of the features from female to male and only then refine those mel features.
>
> 3. So, with the original encoder our diffusion models had to perform two tasks: either refining mel features when conditioned on frequent class label or one-to-one cross-gender voice conversion followed by refining mel features when conditioned on rare class label. We think different nature of these two tasks prevented our models to cope with both of them by just conditioning on a class (in this case also a task) label. This is why we chose an alternative encoder that produced "average" mel features. These features do not correspond to any particular voice, but represent some "average" speaker-independent gender-indepent robotic voice. With such an encoder, models had to perform voice conversion from this "average" voice to the target one and further refine mel features for good sound quality for both speakers. In this setting the tasks for two speakers are of similar nature, and both LIMs and DMs coped with them satisfactorily for both speakers, although with different quality.
>
> 4. In our text-to-speech experiments we use ODE sampler following the original paper describing diffusion-based Grad-TTS model. Similarly to the authors of the latter, we found that ODE sampler leads to better results than stochastic methods of sampling (both the proposed SDE-E and the conventional SDE-A) when small NFE is used. We think that this is because there is a very strong conditioning in this task (i.e. text to be pronounced) and too much uncontrolled variation in the generated voice coming from stochasticity at each generation step harms the overall naturalness of the synthesized speech. As for SDE-E and SDE-A, we did not found any noticeable difference between them in these text-to-speech experiments both in small and large NFE regimes.
>
> [1] Score-based Generative Models with Levy Processes, Yoon et al.

---

> ### Author Response · Authors · 2024-11-19
> **Reply to Q3**
>
> We've additionally ran 4 generation experiments according to the review of the Reviewer EF2f for more reliable results on imbalanced CIFAR10 (see item 5 in the general comment above), so Table 4 has been updated, but it still reflects that ODE-based sampling is not substantially worse than SDE-based for less represented modes. Our explanation of this phenomenon is based on approximate optimal transport properties of deterministic sampling from diffusion models.
>
> As shown in [2], common Variance Preserving Gaussian diffusion models perform nearly optimal transport between prior Gaussian noise and data distribution when ODE-based sampling is employed. Intuitively it means that two points from the prior noise that are close to each other are mapped to close data points under ODE sampling dynamics, and vice versa: far noise samples are mapped to far data points. Thus, data outliers and rare modes are obtained by ODE sampling starting from noise samples belonging to the tail of the prior distribution. Although it was shown in [3] that optimal transport property does not hold precisely for common diffusion models, experimental results in [2] provide convincing evidence that it holds approximately and can be used, for example, to perform style transfer.
>
> If this property also held for LIMs with $\alpha$-stable prior, it would explain the phenomenon you mention: when starting random noise is sampled from the tail of $\alpha$-stable prior, ODE-based sampling results in data sample from a less represented mode. So, even though ODE-based inference does not allow for jumps at sampling, rare classes generation could still be possible due to heavy tails of $\alpha$-stable prior and optimal transport property.
>
> [2] Understanding DDPM Latent Codes Through Optimal Transport, Khrulkov et al.
>
> [3] The flow map of the Fokker-Planck equation does not provide optimal transport, Lavenant & Santambrogio

---

> > ### Comment · Reviewer_7oeL · 2024-11-26
> > **Final comment**
> >
> > **Q1**: I would like to thank the author for their explanation on the sources of errors and exactly why their sampling method is superior. I also think that the added reference directly to the relevant equations in Yoon et al. adds clarity to the paper. If I understand correctly, the difference between your process and Yoon et al.'s process is that you both have the same marginal at each time step but the distribution on trajectories/diffusion paths are different.
> >
> > **Q2**:
> > 1. Thank you for the insight. I guess exploring disentanglement of trajectories might be an interesting topic to explore on toy models, also increasing the dimension of the problem (as it might be stronger in larger dimensions). Although this is beyond the work of this paper, and might also not be worthy of a publication in itself.
> >
> > 2. and 3. Thank you for the details and interpretation on the role of the encoder and the structure of the latent space. I understand better why this setting is necessary and why the comparison between LIM and DM is fair.
> >
> > 3. (Adressed above)
> >
> > 4. Strong conditionning with an almost "collapsed" target distribution compared to the general distribution of speech would indeed be a good explanation for this phenomenon.
> >
> > **Q3**: Part of my initial comment/question (ODE seemingly favorable in the less frequent classes) has been voided by the decreased variance in Table 4 (author's general comment, section 5). I appreciate the authors re-running these experiments showing my comment was wrong. ODE sampling is indeed not substantially worse than SDE based sampling, with moderate drops in FID and coverage. It also does not seem to be specifically favorable to either frequent or less frequent classes (contrary to my previous observation).
> >
> > I also appreciate the author's insightful explanation on the OT approximation on why jumps might not be crucial to rare mode sampling.
> >
> > **Conclusion** Given the authors' answers and modifications, I maintain my score of an 8, as it's a good paper, clearly above the acceptance threshold. Good luck.

---

### Author Response · Authors · 2024-11-19
**General comment and updates summary**

We would like to thank all the reviewers for their valuable comments. We have updated our paper according to some of their suggestions, and here's a brief summary of these updates:

1) We have added a reference to the formula in the paper [1] where the intractable process $Z_{t}$ is defined through the expression of its characteristic exponent as well as a brief comment on it following the suggestion of the Reviewer fdtf who marked this process as unclear and not properly explained.
2) In the section devoted to experimental results we have made the notation more clear and easy to follow and the results easier to compare by replacing ODE(10), SDE(9) and SDE(11) with ODE, SDE-A (approximate SDE) and SDE-E (exact SDE, our method) correspondingly as suggested by the Reviewer 9Gsh. These notation changes also affected Tables 1-4 and Figure 4.
3) Addressing the weakness pointed out by the Reviewer 9Gsh, we have added more discussion on the results from Table 4.
4) We have added FID score obtained by common diffusion model on imbalanced CIFAR10 to serve as a reference.
5) Addressing concerns of the Reviewer EF2f who pointed out that the improvement on imbalanced CIFAR10 is very little, we performed 5 runs of generation on this dataset instead of 1 to obtain more reliable results and updated Tables 3 and 4. They did not change significantly, the proposed method SDE-E still outperforms SDE-A (0.3 FID and 1.5% coverage). In Table 4 coverage for less represented classes (with larger class ids) changed to more extent after 4 additional experiment runs than that for more represented classes because
coverage is calculated as a percentage of images within a class having at least one generated image as their nearest neighbour, and the number of images in less represented classes is very little (50 for the smallest class with id=9) meaning that variance of the results across different runs is larger.

[1] Score-based Generative Models with Levy Processes, Yoon et al.

---

### Meta-Review · Area_Chair_e2jz · 2024-12-21

**Metareview:**

This paper proposed a parametric family of SDEs for Lévy-Itô diffusion models to improve the sampling accuracy and efficiency. The proposed method is verified through experiments on imbalanced datasets for different tasks.

Strengths:
1. The proposed SDE formulation is novel and mathematically rigorous, addressing specific limitations of prior Lévy-Itô diffusion models.

2. The method has good flexibility and can be applied to different tasks such as image generation and speech synthesis.

3. The method shows consistent improvements in sample quality, and robustness in imbalanced datasets.



Weaknesses:

1. The gains in imbalanced datasets are not very big. Nevertheless, they are consistent and meaningful.

2. The comparisons with SOTA methods could be more extensive.

Overall, the paper presents a interesting theoretical contribution with empirical success in  different applications. While some concerns regarding the gains and  baseline comparisons remain, considering the overall quality and novelty of the work, I would like to recommend acceptance.

**Additional Comments On Reviewer Discussion:**

The discussion phase highlighted key points, most of which were addressed effectively by the authors, e.g., improvements in imbalanced datasets, relationship between the proposed method and prior work, more diverse datasets and baselines.


The authors demonstrated a strong effort during the rebuttal phase, addressing most reviewer concerns.

---

### Decision · Program_Chairs · 2025-01-22

Accept (Poster)